# Alternate Wetting and Drying in the Center of Portugal: Effects on Water and Rice Productivity and Contribution to Development

**DOI:** 10.3390/s22103632

**Published:** 2022-05-10

**Authors:** José Manuel Gonçalves, Manuel Nunes, Susana Ferreira, António Jordão, José Paixão, Rui Eugénio, António Russo, Henrique Damásio, Isabel Maria Duarte, Kiril Bahcevandziev

**Affiliations:** 1Instituto Politécnico de Coimbra, Escola Superior Agrária de Coimbra, 3045-601 Coimbra, Portugal; mnunes@esac.pt (M.N.); susana.ferreira@esac.pt (S.F.); iduarte@esac.pt (I.M.D.); kiril@esac.pt (K.B.); 2Direção Regional de Agricultura e Pescas do Centro, 3000-317 Coimbra, Portugal; antonio.jordao@drapc.gov.pt; 3Associação de Beneficiários da Obra de Fomento Hidroagrícola do Baixo-Mondego, 3140-901 Montemor-o-Velho, Portugal; jmjpaixao@gmail.com (J.P.); dilarusso@gmail.com (A.R.); 4Associação de Regantes e Beneficiários do Vale do Lis, Monte Real, 2425-000 Leiria, Portugal; eugenio-rui@sapo.pt (R.E.); hdamasio71@gmail.com (H.D.); 5CERNAS—Research Centre for Natural Resources, Environment and Society, 3045-601 Coimbra, Portugal

**Keywords:** rice irrigation, *Oryza sativa* L., water saving, AWD, MEDWATERICE, Lis Valley, Lower Mondego, Portugal

## Abstract

Rice irrigation by continuous flooding is highly water demanding in comparison with most methods applied in the irrigation of other crops, due to a significant deep percolation and surface drainage of paddies. The pollution of water resources and methane emissions are other environmental problems of rice agroecosystems, which require effective agronomic changes to safeguard its sustainable production. To contribute to this solution, an experimental study of alternate wetting and drying flooding (AWD) was carried out in the Center of Portugal in farmer’s paddies, using the methodology of field irrigation evaluation. The AWD results showed that there is a relevant potential to save about 10% of irrigation water with a reduced yield impact, allowing an additional period of about 10 to 29 days of dry soil. The guidelines to promote the on-farm scale AWD automation were outlined, integrating multiple data sources, to get a safe control of soil water and crop productivity. The conclusions point out the advantages of a significant change in the irrigation procedures, the use of water level sensors to assess the right irrigation scheduling to manage the soil deficit and the mild crop stress during the dry periods, and the development of paddy irrigation supplies, to allow a safe and smart AWD.

## 1. Introduction

Rice (*Oryza sativa* L.) is the worldwide major staple crop, cultivated in over 164 Mha [1] and essential for ensuring global food security, given that over 90% of production is used directly for human consumption. On the other hand, rice is a very high-water demanding crop, making the water resources a limiting factor for sustainable production. This issue is compounded by the fact that the world demand for rice is increasing due to the population increase in the regions that consume more and its high nutritional content. Rice crop has an important economic and social value in several regions, namely in Mediterranean countries [1]. In Portugal, rice is cultivated in about 30 thousand ha, especially in the Mondego, Tagus, and Sado Valleys, in lowland areas and coastal wetlands, with a particular role in the preservation of biodiversity and soil conservation [2].

Rice is cultivated in paddies and traditionally has been irrigated by continuous flooding (CF) for environmental and microclimatic reasons for several centuries. The main problems related with flooding rice irrigation refer to a high-water demand, nutrients lost by leaching and runoff, soil methane emissions to the atmosphere, and the agrochemical pollution of water resources of agroecosystems [3]. Facing the increasing threat of water scarcity, it urges developing agronomic and irrigation practices to reduce water use, while maintaining or increasing land and water productivity. In short, the efforts for the sustainability of rice crop are of strategic importance in the context of food security. A great research effort has been made in the last decades to answer these challenging questions, looking for alternatives to CF, which is a key element to water saving and safeguarding the environmental quality of rice agroecosystems [4,5]. Numerous studies have demonstrated agronomic and irrigation advances, namely focusing on land preparation through precise land leveling (PLL) and soil tillage [6]; multiple irrigation system solutions, such as zero-grade fields, and alternate wetting and drying [7,8]; multiple-inlets and furrows [9]; pivot irrigation [10]; drip irrigation [6,9,10,11,12]; and advances in cultivation pattern, such as the system of rice intensification, direct seeding methods cultivation, aerobic rice systems, ground cover rice production systems, and genetic approaches [6,13].

Alternate wetting and drying irrigation (AWD) consists of intermittent flooding through a sequence of flooding cycles with very thin water depths (about 5 cm), followed by drying periods. The recession is only due to infiltration and evaporation, leaving the soil surface layer in a non-saturated condition for a few days (a condition called “dry soil”, in contrast to “flooded soil”) until the next reflooding cycle [14,15]. The soil is kept dry until hairline cracks are visible, or the decrease in the soil water potential does not cause significant crop stress. To avoid compromising production, the AWD must consider the thermoregulatory effect of the water, in the most critical stages of the crop (such as panicle differentiation, flowering, and early ripening), ensure weed control, and the protective effect against strong winds. AWD has been successfully used in several countries, such as India [16], Bangladesh [17], Philippines [18,19], Vietnam [20], China [21], and the USA [5,22].

The benefits of AWD, when compared with CF, include: (i) irrigation water savings by up to 30% [15,23,24] due to the decrease in deep percolation, facing a lower soil water pressure, and a decrease in the soil evaporation; (ii) a reduction of greenhouse gas emissions (methane plus nitrous oxide) by 45–90% [3,25]; (iii) a reduction of the arsenic accumulation in the grain by 50% [26,27,28,29]; and (iv) a reduction of methylmercury concentrations in rice grain by 38–60% [30,31] and in the soil [30].

The AWD management is based on two parameters: timing and threshold [24,32]. The timing is when in the growing season the drying cycles are imposed, namely by the vegetative, reproductive, or ripening phases, or then throughout the crop season. The crop sensitivity to water stress is a major factor to determine this timing. The AWD threshold is the value of a soil water content that refers a limit condition of water deficit used to determine the time for reflood. There are two categories for AWD threshold (Table 1): (i) Severe, implying a high risk of crop water stress, even a significant yield reduction, usually when the soil water potential drops below −20 kPa, corresponding to the time after ponded water disappeared (TAD) higher than seven days; and (ii) Mild (or safe), when the soil water potential (SWP) is not outdated (SWP ≥ −20 kPa), or the field water level (FWL) is not allowed to drop more than 15 cm below the soil surface (FWL ≤ 15 cm), corresponding a TAD between 5 and 7 days.

The irrigation systems of paddies can benefit from the advances that the automation of surface irrigation have undergone, due to a great effort of modernization on gate control, remote and feedback control, and real-time optimization [33,34,35,36,37]. For example, Masseroni et al. [36,37] presented the first automatic system for CF paddy irrigation in Europe. Automation aimed at AWD is more demanding than in CF, requiring articulating sensor data for a reliable control process to avoid the risks of crop water stress. AWD alters the water management logic of CF; namely, the management process, comprising the use of the following equipment and sensors [38]: (i) automatic devices to control paddies water supply, aiming to save labor and for water accounting to collect data for irrigation management; (ii) field water level (FWL) sensors, placed on water tubes, to continuously measure the water level above and below the soil surface; (iii) soil water potential (SWP) sensors placed in the rooting zone, to monitor the matrix tension to assess the conditions of crop water stress, and providing data to manage the irrigation scheduling; (iv) automatic weather stations, proving real-time climatic data to measure the rainfall and determine the crop evapotranspiration and irrigation water requirements; and (v) the integration of information through telemetry and digital systems, allowing the precise control of irrigation managing, using the feedback of several of the sensors data, used automatically to control and avoid the risk of crop water stress due to the drying periods of AWD, in a framework of labor reductions [36]. These smart and automatic solutions have a high potential for practical application, namely at a district scale [38]; however, there is a lack of required information on protocols for optimizing AWD, incorporating information on specific crop water stress, as well as farm commercial solutions.

This research aimed to provide knowledge to outline the guidelines to promote the development and automation of AWD by rice farmers, by studying the effects of AWD on rice yield and water use relative to the actual practice of CF in the Central Region of Portugal.

## 2. Materials and Methods

The experimental study was carried out in 2020 in the Lower-Mondego and Lis Valley Irrigation Districts, located in Coastal Center of Portugal, with a total irrigated area of about 14,000 ha, and a rice area of about 6000 ha [39,40] (Figure 1a). This region has a Mediterranean climate, Csb and Csa of Köppen classification, with an annual average precipitation of about 800 mm to 900 mm. It has temperate and mild summers, with virtually no rainfall, and rainy winters with mild temperatures [41] (Figure 2). The soils are mainly alluvial with high agricultural quality, some of which are poorly drained, with waterlogging and salinization risks, particularly on the downstream areas where rice is cultivated in paddies [39]. The river water used for irrigation is diverted and conveyed mainly by gravity, from weirs, through a collective system managed by Water User’s Associations (WUA) [39].

In these valleys, rice is cultivated in traditional paddies, on lower soils with heavy texture and poor drainage, with a shallow and relatively saline groundwater table. Paddies are irrigated by CF, with ca. 10 cm of ponding depth, and a frequency varying from daily to a few days. The flooding of paddies plays several determinant roles [42]: (i) temperature regulation during the first weeks of crop development due to microclimatic imperatives, particularly during night-time in the initial phase of the cycle and during flowering; (ii) after sowing, to avoid seed collecting by wild birds; (iii) control of weeds development; (iv) control of the crop damage due to the strong wind; and (v) soil salt leaching in susceptible areas. In its turn, the initial drainage periods enable for example: (i) the application of phytopharmaceuticals, especially herbicides and fungicides; and (ii) a good rooting of the seedlings, while avoiding soil hardening, and a reduction of algal proliferation on surface water. The paddies are highly water demanding due to a significant deep percolation, and surface drainage [42].

The experimental design, at each site, consisted of two rice plots located in identical edaphoclimatic conditions, one irrigated by CF and the other by AWD. Three trial sites were selected: Bico-da-Barca (BB) and Quinta-do-Canal (QC) in the Lower-Mondego, and Nuno-Guilherme (NG) in the Lis Valley, mapped in Figure 1b, being their geographic coordinates and soil characteristics presented in Table 2.

A single Italian rice cultivar, Ariete (japonica type) was used in all the sites. Ariete is classified as semi-early, with a cycle of about 139–150 days. It was sown in mid-May, and harvested throughout October, and was fertilized with doses of about 70–90 kg N/ha. Crop development and irrigation practices, and corresponding dates, are presented on Table 3 (example of NG site).

The experimental plots with the CF treatments were fully managed by the farmers. Traditional flooding practices were applied, which were used as a reference to compare with the AWD. Identical agronomic practices were adopted in both treatments, namely the soil preparation, including the ploughing and harrowing, land levelling, fertilization, wet sowing, and crop protection treatments. Water from the river was supplied by gravity-fed systems, using open canals and buried pipes, which were manually controlled.

The methodology adopted in the AWD plots was based on the description by Bouman et al. [23], in the framework of the Mild version, with adjustments, according to the local experimental conditions. In summary, the following steps were taken: (i) an initial flooding for wet sowing, followed by an initial drying through a fast surface drainage event, to favor rice emergence like the traditional practice; (ii) shallow ponding during the vegetative phase, considering the drying periods required for herbicide application, usually twice, like the traditional practice; (iii) AWD technique applied after the vegetative phase, taken in account that: (a) the target was a flood water depth not higher than 5–7 cm; (b) the irrigation schedule considered was an interval of 10 to 14 days between irrigation events; (c) the water level should not fall to 15 cm below the soil surface, measured in a water tube; (d) Particular attention was paid on the flowering period because at this phase plants are very sensitive to water stress; and (iv) the last irrigation event took place about 20 days before the harvest.

The hydraulic monitoring system installed had two components: water tubes with automatic sensors, and water accounting devices with continuous record. The water tubes, consisting of PVC pipes, were placed on soil at 25 cm depth. These tubes, were 40 cm long and 10 cm in diameter, have holes with 1 cm in diameter through which the soil water flows into its lumen, allowing the observation of FWL and the measurement with a piezometric head. Figure 3 shows a water tube installed in a rice plot, during a dry period of AWD. The water tubes were equipped with automatic water level sensors, where data were complemented with the measurement of the atmospheric pressure through a barometer located nearby (Table 4). Regularly, at least once a month, the data from loggers were downloaded to a PC for further data analysis. During the crop season, manual FWL measurements were carried out in the water tubes with a ruler, and the data were used for testing and calibrating the sensors.

The comparison of CF with the AWD practices was based on the water level recorded on water tubes, elucidating about the water level above the soil surface in the flooding irrigation plots, during the entire crop season.

The measurement of the paddies inflow and the outflow discharges resorted to flumes or weirs, in which reference water head was measured by automatic water level sensors (Table 4) [45,46]. In its turn, the calibration of these devices was carried out through the canal section velocity method, where the point velocities were measured with an electromagnetic current meter (Table 4).

The meteorological observations were carried out with automatic weather stations installed near the experimental sites, with a set of sensors for air temperature and humidity, solar radiation, and wind speed, a Class A pan evaporimeter, and remote communication tool via GSM to the several data users (Table 4). Daily reference evapotranspiration was calculated by Penman–Monteith method, based on Allen et al. procedure [47]. The daily crop evapotranspiration (ETc) was calculated through the crop coefficients of 1.25 for flooding condition, and 1.10 for dry periods [47].

These measurements allowed obtaining daily data from the system, necessary for the daily water balance method that enabled to calculate the deep percolation (DP), by applying the Equation (1),
DP = P + I − ETc − SD − ΔSW,(1)
which requires the values of precipitation (P), irrigation (I), surface drainage (SD), and storage difference of surface or subsurface soil water (ΔSW) [42].

The crop yield parameters were determined at harvest, collecting the aerial part of the total rice plants in diverse unit areas of 0.5 m^2^, with about 5-unit areas per hectare. The biomass harvest was latter processed in the laboratory, determining the dry matter of grain with 14% of humidity and straw and the weight of 1000 grains.

Based on the irrigation water applied (I, m^3^ ha^−1^), precipitation (P, m^3^ ha^−1^) and yield (Y, kg ha^−1^), the water productivity (WP, kg m^−3^) was calculated through the Equation (2),
WP = Y/(P + I).(2)

The effect size (ES, %) [24] was calculated, to compare the effects of AWD with several literature data sources, through the Equation (3),
ES_x_ = ln (x_AWD_/x_CF_) × 100%,(3)
where, x is the response variable (yield, water use, and water productivity).

## 3. Results

### 3.1. Soil Flooding Changes and Crop Development

The characterization of the traditional CF practice, illustrated in Figure 4 with data from the NG site, evidenced the contrast with the dry periods in the AWD treatment. It considered four dry periods during the vegetative phase, and that after the 16 July, the AWD technique had been applied, making five wet–dry cycles, until the final period of 20 days before the harvest. These cycles corresponded to a period between 12 and 14 days, with irrigation depths between 92 and 109 mm, and 5 or 6 days, with dry soil, per cycle.

The increase in time with dry soil due to AWD, in contrast with CF, was 10, 25, and 29 days, corresponding to a period with dry soil relative to the cultural cycle of 47%, 57%, and 35%, for QC, BB, and NG, respectively (Table 5).

The irrigation allocations recorded in the CF plots were by decreasing order: 1725 mm in BB, 1588 mm in QC, and 1292 mm in NG (Table 6). These results correspond to AWD relative water savings of 11.8%, 12.6%, and 9.5%, reductions in cultural evapotranspiration of 3.4%, 1.6%, and 2.7%, and reductions in deep percolation of 15.0%, 22.1%, and 13.9%, for BB, QC, and NG, respectively. These percentages are higher if considering “after vegetative phase” only.

Average rice yield (grain with 14% of moisture) was higher in the plots irrigated by CF than in those with AWD. In CF plots, rice production was 9.58 t/ha in QC, 8.10 t/ha in BB, and 5.99 t/ha in NG, whereas in the AWD plots it decreased by 3.4% and 5.6% in QC and NG, respectively, and increased by 0.3% in BB. In turn, the water productivity increased in the three sites, 10.6%, 13.6%, and 3.7%, in the same order, having been the maximum at QC with 0.667 kg/m^3^, 0.543 kg/m^3^ at BB and 0.452 kg/m^3^, in NG. However, the yield varied significantly between the sites, possibly due to the local edaphoclimatic conditions (Table 7).

As air temperature is a determining factor for rice cultivation, namely the average temperature together with thermal variations and critical temperatures [43], the thermic comfort for rice in the plots was assessed. Considering the NG site (Figure 5), the minimum temperatures were higher than 10 °C, and in a short period between the 15 and 16 July, maximum temperatures were 37 °C and 38 °C. From sowing (14 May) to the harvest (10 October), the average air temperature was mild, and the temperature range were within the critical values, except in the reproductive and early maturation phases, when the minimum values were below the critical value (Figure 5).

The temperatures below 17 °C for long periods during the reproductive and the early maturation phases, as observed in the NG site (Figure 5), might explain the lower yield in both treatments at the NG site, in comparison with the yield of the Lower Mondego sites. The average and minimum temperatures recorded were below the optimal, and in some days, below the minimum critical value of 15 °C during the panicle differentiation, which might have delayed flowering and inhibited fertilization [42,49,50].

### 3.2. Outline of Guidelines to Promote AWD Automation

The effective implementation of AWD requires changes in the water supply system and in the irrigation management. The monitoring of the surface and subsurface water level requires real-time information that should be collected by sensors to minimise the water crop stress. The frequent flooding irrigation must be carried out using the automatic gates control, in a logic of irrigation modernization. This technological process is fundamental for the progress of AWD, to reduce the negative impacts of CF.

Based on our experience and on Siddiqui et al. [38], Masseroni et al. [37], and Pham et al. [20], the data required to develop a system for the automation of on-farm scale AWD were identified and organized (Table 8). These data comprise general information about the crop, information adjusted to local conditions, and the incorporation of experience and knowledge for the rational management of AWD.

The control unit integrates information and, through the soil water balance, allows the assessment of the system performance, and to get irrigation decisions, to open the gates during a specific time, or to close them. The feedback loops are fundamental to a safe control of soil water and irrigation inflow.

The logic of the information transmission process is presented in the flowchart of Figure 6, including the data collected by the farmer, and automatic sensors, to the irrigation management decision, articulated with the water supply conditions at district level [7,51,52].

## 4. Discussion

The analysis of the observed effects of AWD on rice yield, water use, and water productivity, according to the Equation (3) (Table 9), show that the unique value that contradicts the expected tendency of AWD, is the effect on yield at the BB site, which did not decrease in the AWD treatment in comparison with the CF one. The comparison of the effects of AWD on yield, water use, and water productivity with the published results of worldwide studies [24], is presented in Figure 7 and Figure 8. It was observed that the overall effect of AWD (Figure 7): (i) on yield—the values of QC and NG sites are both negative (−3.5% and −5.7%), unlike those from BB, which has an effect very close to zero; (ii) on water use—the observed effects are significantly different (about −11%, compared with −26%); and (iii) on water productivity—the observed effects are also significantly different (3.6 to 12.8%, compared with 24%). The first conclusion of this comparison is that the applied AWD had a lower impact on yield, and a lower impact on water saving and water productivity.

The observed effects of AWD on yield were compared with results presented by Carrijo et al. [24], the relative to Mild AWD option, which was the practice adopted in our study. The observed effects relative to the soil properties influencing AWD (pH, SOC, and texture), are very close to the Carrijo et al. [24] ones obtained worldwide, showing that the effectiveness of the AWD is dependent on those soil characteristics (Figure 7).

The effect of those soil characteristics on the effectiveness of AWD in terms of crop productivity could be explained by the following reasons, in which relevance is more significant in severe AWD [24,29]: (a) The effects of pH: (i) in alkaline soils the high percentage of exchangeable sodium (Na) damages the soil structure and induces impermeable layers; the soil compaction does not have a negative impact on the CF, as the root system is very superficial, but on the contrary, it affects the rice irrigated with an AWD system as it limits the depth of the roots, which cannot compensate for the lower moisture in the surface soil layers; (ii) the high levels of Na in alkaline soils can lead to Na toxicity effects in plants; this problem is mitigated in the CF in the soil saturation condition, due to the higher Na soil leaching and the higher SWP value; however, in unsaturated soil the higher Na concentration causes a greater toxicity risk, thus, rice becomes less tolerant under AWD, with a lower productivity; and (iii) alkaline soils may have higher nitrogen losses due to ammonia volatilization, which is more prevalent in non-flooded conditions; therefore, lower yields under AWD may be due to reduced N root uptake in alkaline soils. (b) The effect of SOC: (i) the higher soil organic matter positively affects the soil water holding capacity and the plant-available water, due to its direct impacts on lower bulk density, aggregate stability, high porosity, and improved structure; and (ii) the higher soil organic matter positively affects the nitrogen mineralization in aerobic soil conditions, implying that N availability may be increased under AWD systems conducted in high SOC soils. (c) The effects of texture: the higher clay content reduces the soil percolation, and nitrate leaching, implying that higher levels of nitrogen are available to the root system.

AWD could not be used in lowland soils with serious drainage problems, where dry soil conditions are not feasible due to the high level of groundwater table (data not shown). On the other hand, AWD is not recommended in situations of high level of salinization of the very shallow groundwater table, to avoid the capillary rise of salts to rice root zone during the dry periods.

The water savings and the impacts on production due to AWD, recorded in our study are, in general, in agreement with the values indicated in the literature [6,7,16,18]. Therefore, this experiment confirmed the importance of AWD for water saving in rice irrigation, especially from the reproductive phase, after the middle of July onwards. This water saving allows the WUA’s to mitigate the water scarcity in this period at the district level, which corresponds to the maximum demand of most irrigation crops, such as corn, in Portugal [53]. The successful application of AWD also requires several changes in the rice production system, namely on PLL, weed control, and fertilization scheme. PLL is a crucial complementary aspect to the success of AWD, so that the water depth on the soil is uniform throughout the entire plot. This is a condition for adopting a thinner water layer that, therefore, allows for a reduction in the use of water [54]. To this end, a regular and rigorous practice of level maintenance and monitoring should be encouraged [55].

The AWD negative impacts on yield raises the question of the farmer’s economic income, making this technique unattractive, especially when the water supply is sufficient for CF. This issue claims for a political strategy to promote rice production sustainability because the governmental support to change the rice irrigation system should guarantee the farmer’s income. Moreover, the recognition of another positive environmental impact of AWD, with the increasing period that the soil is in aerobic conditions, is the reduction of methane emissions into the atmosphere [56].

Irrigation management in the alternating flooding period can be carried out in several ways. The simplest procedure consists of measuring the level of the saturated soil, following its lowering during the drying period, and establishing the critical level to determine the most opportune moment for the next flooding. Another option is based on measuring the soil water potential in the root zone, being the timing of irrigation based on this parameter. Therefore, the use of soil moisture, or soil water potential sensors to control the crop water stress during the drying phase is an important issue [57]. These different methods can be articulated with sensing mechanisms aimed at automating irrigation [36,52], to optimize water consumption and crop productivity, and reduce the additional manpower required by the AWD. This topic represents an important and current area of investigation, which deserves the best attention [58]. The application of information and communication technologies to develop user-friendly and smart solutions are being tested [20,37,38,59]. Effectively, the issue of rice irrigation automation opens a high opportunity to the hydraulic equipment companies, due to the large number of rice growers, as potential clients. Nonetheless, some research questions should be previously studied.

Here, we present a contribution towards to the AWD automation, recognizing that it is the right pathway for the future of flooding rice irrigation. It should be highlighted the importance of a knowledge base to support designers, extensionists, and farmers in this process.

## 5. Conclusions

This study confirmed the interest of the AWD irrigation of rice paddies in the Center of Portugal, a technique to be applied after the vegetative phase of the crop. On one hand, AWD allows water savings in relation to CF, between 10% and 13%, without significantly compromising production, and an increase in water productivity between 3.7% and 13.6%. This saving occurs from the reproductive phase to the end of the season, therefore, during the period of the greatest water demand to irrigate most crops. On the other hand, AWD makes it possible to significantly increase the number of days with non-flooded soil, between 10 and 29 days, with a consequent reduction in methane emissions into the atmosphere (not assessed in this study).

The practice of waterlogging in the early stages of the crop is highly conditioned by particularly sensitive agronomic criteria (thermal control, weeds, wind, and phytosanitary treatments). Therefore, changes of the conventional procedure are not recommended until the beginning of the reproductive stage. Furthermore, the need to carry out frequent and planned irrigation events during the AWD period, demands for more accurate inflow control devices, making place for its automation, and leading towards rice modernization through smart flooding irrigation systems.

To our knowledge, this is the first report of a study on rice irrigation with AWD in Portugal. This work contributes a local characterization of AWD effects on water and rice productivity on paddies in Portugal. Field assessment is crucial to manage the water use in the agroecosystem to control the long-term negative environmental and public health impacts.

## Figures and Tables

**Figure 1 sensors-22-03632-f001:**
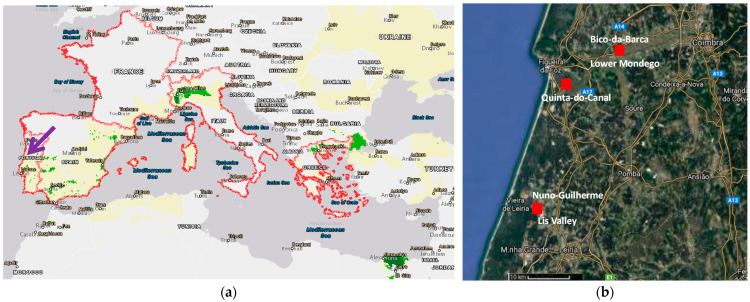
Geographic location of: (**a**) Mediterranean rice growing areas, limited by a red line with respective production: (

), 101–1000 tons, (

), 1001–10,000 tons, and the study area in Central Costal Region of Portugal (arrow); (**b**) the experimental fields (
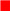
), on the Lower-Mondego and Lis Valleys (source: (**a**) [40]; (b) Google Maps, https://maps.google.pt (accessed on 20 March 2022)).

**Figure 2 sensors-22-03632-f002:**
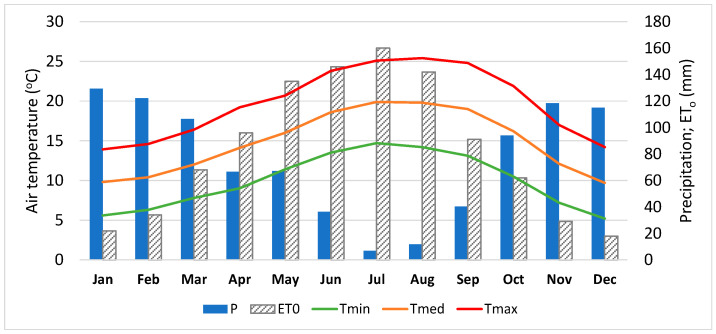
Average monthly air temperature, precipitation, and reference evapotranspiration of the Lower Mondego study area, relative to the 1981–2021 period (P—precipitation; ET0—reference evapotranspiration; Tmin—minimum air temperature; Tmed—medium air temperature; Tmax—maximum air temperature) (source: [41]).

**Figure 3 sensors-22-03632-f003:**
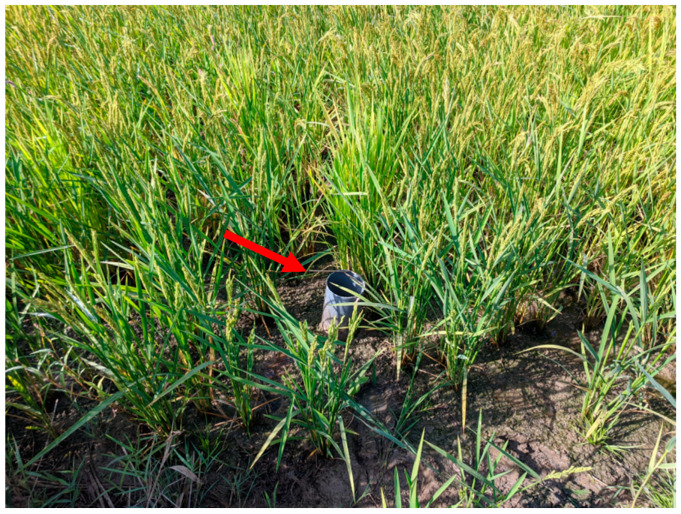
A water tube (red arrow) installed in a rice plot, during a dry period of AWD.

**Figure 4 sensors-22-03632-f004:**
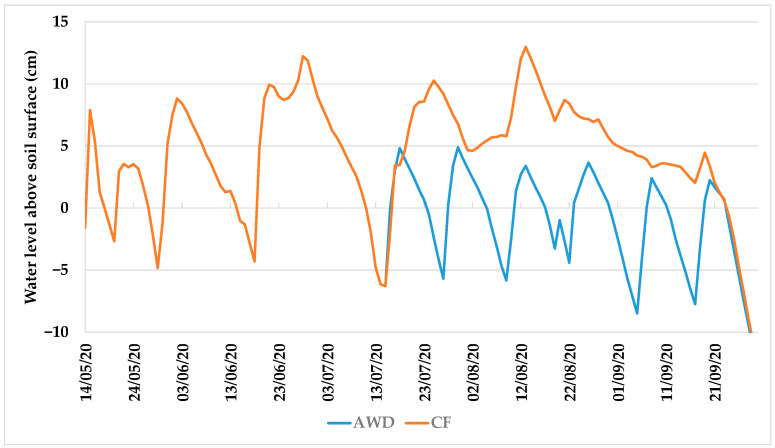
Water level above and below the soil surface (cm) of CF and AWD irrigation during 2020 rice crop season in NG site, Lis Valley (CF—continuous flooding irrigation; AWD—alternate wetting and drying irrigation).

**Figure 5 sensors-22-03632-f005:**
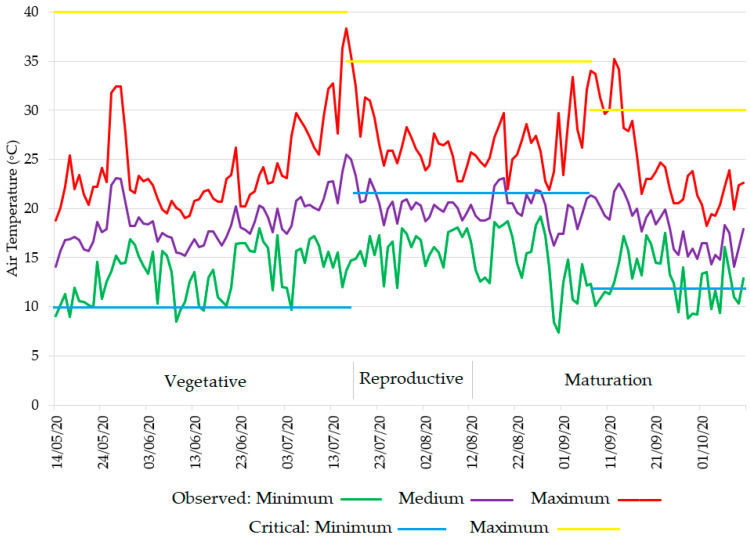
Daily air temperatures (minimum, average, and maximum) during 2020 rice crop season in NG plot, Lis Valley. The critical minimum and maximum temperatures were adapted from Yoshida [49], Krishnan et al. [50], and Pereira [42].

**Figure 6 sensors-22-03632-f006:**
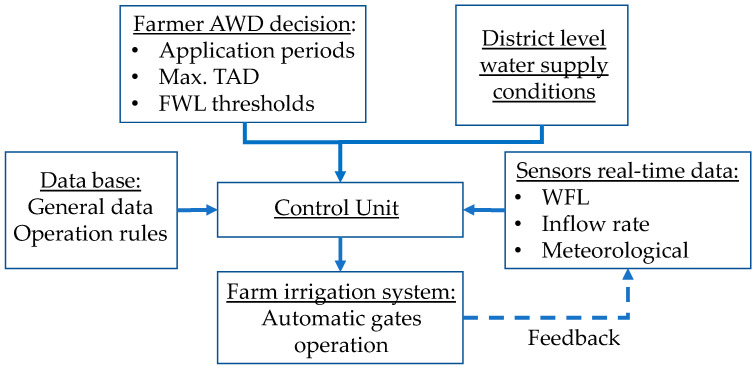
Flowchart of the logic of the automatic AWD system.

**Figure 7 sensors-22-03632-f007:**
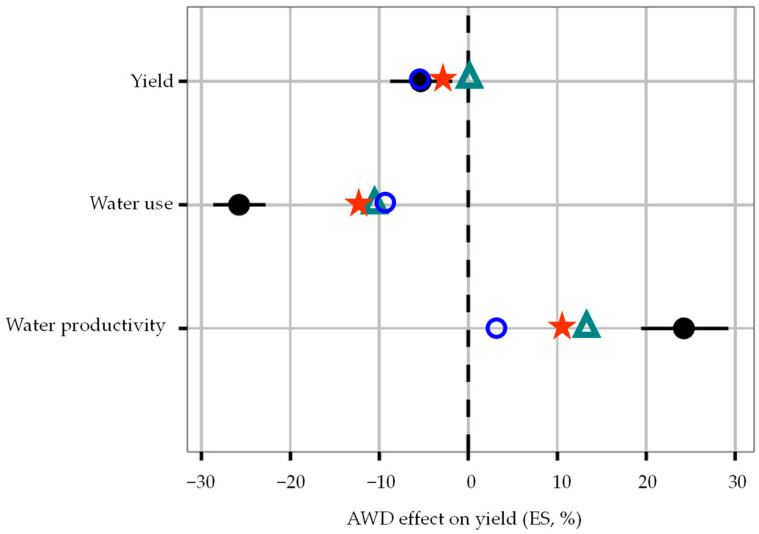
Overall effect of AWD on yield, water use, and water productivity. Legend: Data presented by Carrijo et al. [24]—mean and confidence interval, 
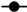
; Quinta-do-Canal, QC—
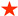
; Bico-da-Barca, BB—
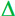
; Nuno Guilherme, NG—
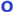
. Graphs adapted from Figure 1 of Carrijo et al. [24].

**Figure 8 sensors-22-03632-f008:**
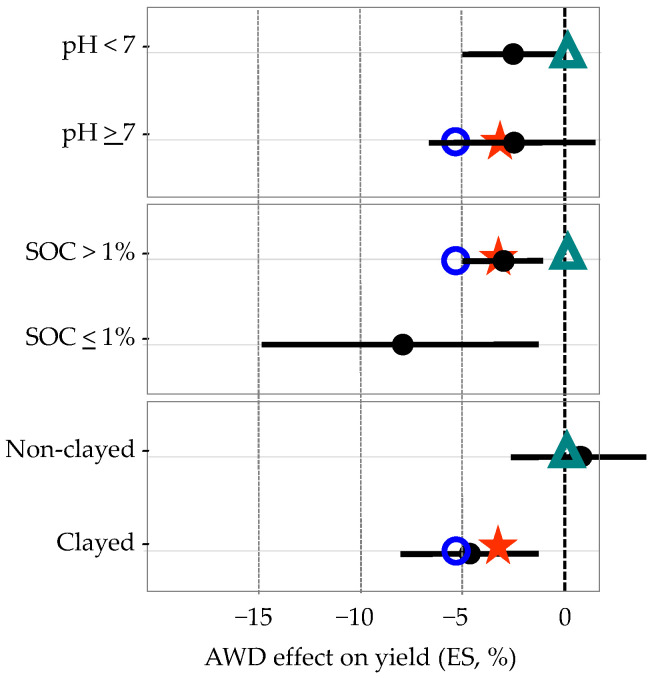
Effect of AWD under Mild AWD (ES, %), according to the soil variables: (a) pH; (b) Soil organic carbon (SOC); and (c) soil texture. Legend: Data presented by Carrijo et al. [24]—mean and confidence interval, 
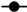
; Quinta-do-Canal, QC—
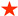
; Bico-da-Barca, BB—
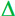
; Nuno Guilherme, NG—
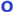
. Graphs adapted from Figure 4 of Carrijo et al. [24].

**Table 1 sensors-22-03632-t001:** AWD management parameters and thresholds.

Parameter	Units	AWD Threshold		Data Source
		Severe	Mild	
Time after ponded water disappeared (TAD)	days	>7	5–7	[15,32]
Field water level (FWL)	cm	>15	<15	[15,24,32]
Soil water potential in the rooting zone (SWP)	kPa	<−20	<−10 to −15	[23,24]

**Table 2 sensors-22-03632-t002:** Study site characteristics.

Characteristics	Parameters	Experimental Sites		
		BB	QC	NG
Location	Latitude	40°10′31″ N	40°06′54″ N	39°52′17″ N
	Longitude	8°39′40″ W	8°48′08″ W	8°52′58″ W
	Altitude (m)	5	2	8
Type of farm		State agricultural experimental station	Associative	Private
Area (ha)	Field plots	0.11	4.8	3.0
Texture (%)	Sand	30.0	6.4	7.1
	Silt	49.3	59.2	37.3
	Clay	20.7	34.4	55.6
Texture class *		Silt loam	Silty clay loam	Clay loam
Soil	pH (H_2_O)	5.9	7.6	7.2
	Soil Organic Carbon (%)	2.3	2.3	2.7
	Bulk Density (g cm^−3^)	1.28	1.28	1.25
Groundwater table level (bss, cm)		40–80	50–80	75–85
Soil Water Content (cm^3^ cm^−3^)	Saturation	0.519	0.517	0.520
	Field Capacity	0.484	0.471	0.385
	Wilting Point	0.090	0.188	0.204

* Texture classification according to Gomes and Silva [43]; Soil characteristics are relative to the superficial depth of 60 cm; bss—below the soil surface. Experimental sites: BB, Bico-da-Barca; QC, Quinta-do-Canal; and NG, Nuno-Guilherme (data source of soil texture (% and class) was adapted from [44]).

**Table 3 sensors-22-03632-t003:** Crop development and irrigation practices and corresponding dates (NG site data).

Crop Development and Irrigation Practices	Days after Sowing *, DAS
Initial soil flooding	−1
Wet sowing	0
Start tillering	34
Panicle differentiation	60
Start AWD	67
Flowering	90
Last irrigation event	128
Harvest	148

* Sowing on 14 May 2020.

**Table 4 sensors-22-03632-t004:** Sensor characteristics applied in the experimentation.

Objective	Brand and Model
Water level	In-Situ Inc., model Rugged TROLL 100, Fort Collins, CO, USA
Atmospheric pressure	In-Situ Inc., model Rugged Baro TROLL, Fort Collins, USA
Water flow velocity	VALEPORT, EM flow meter model 801 flat, Decon, UK
Pluviometer	Pronamic ApS, diam. 16 cm, Ringkøbing, Denmark
Data logger	Campbell Scientific, Inc. CR300, Logan, UT, USA
Air temperature and humidity	Campbell Scientific, Inc. EE181, Logan, UT, USA
Solar radiation	Campbell Scientific, Inc. CS301, Logan, UT, USA
Wind speed	Lambrecht meteo GmbH, ORA, Göttingen, Germany
Remote communication of weather station	Cinterion, BGS2 Terminal RS232, Praha, Czech Republic

**Table 5 sensors-22-03632-t005:** Number of days with wet and dry soil, in the experimental rice fields irrigated with CF and with AWD.

Experimental Site	Soil Condition	Crop Season (Days)		After Vegetative Phase (Days)	
		CF	AWD	CF	AWD
QC	Wet	88	78	40	29
	Dry	59	69	33	44
	Total	147	147	73	73
BB	Wet	83	58	40	21
	Dry	52	77	39	58
	Total	135	135	79	79
NG	Wet	118	89	68	39
	Dry	22	51	7	36
	Total	140	140	75	75

CF—Continuous flooding; AWD—Alternate Wetting and Drying; Experimental sites: BB, Bico-da-Barca; QC, Quinta-do-Canal; and NG, Nuno-Guilherme. Adapted with permission from Oliveira et al. [48], 2022, IGI Global.

**Table 6 sensors-22-03632-t006:** Water use parameters in the experimental rice fields irrigated with CF and with AWD, during the crop season, and after the vegetative phase.

Experimental Site	Water Use (mm)	Entire Cropping Season		After Vegetative Phase	
		CF	AWD	CF	AWD
QC	ETc	696.3	685.1	298.9	287.7
	I	1588	1388	651.5	425.1
	P	130.4	130.4	77.6	77.6
	DP	538.5	419.7	261.4	152.6
	SD	516.4	460.8	211.6	117.0
BB	ETc	588.0	568.1	282.0	263.4
	I	1725	1522	742.1	537.5
	P	99.6	99.6	87.8	87.8
	DP ^1^	1264	1075	651.4	494.2
NG	ETc	674.7	656.3	347.4	329.0
	I	1292	1169	639.2	517.0
	P	81.8	81.8	68.4	68.4
	DP	587.8	506.2	330.2	248.6
	SD	154.8	118.2	36.5	0

ETc—Crop Evapotranspiration(mm); DP—Deep percolation (mm); P—Precipitation (mm); I—Irrigation (mm); SD—Surface Drainage (mm); CF—Continuous flooding; AWD—Alternate Wetting and Drying; Experimental sites: BB, Bico-da-Barca; QC, Quinta-do-Canal; and NG, Nuno-Guilherme. ^1^ Includes a small fraction of surface drainage. Adapted with permission from Oliveira et al. [48], 2022, IGI Global.

**Table 7 sensors-22-03632-t007:** Rice and water productivity of CF and AWD plots.

Experimental Site	Method	Y (t/ha)	WP (kg/m^3^)	G (g)	RS (t/ha)
QC	CF	9.582 ± 1.230	0.603	28.9 ± 1.42	5.49 ± 0.70
	AWD	9.252 ± 6.120	0.667	28.9 ± 0.74	5.62 ± 0.53
BB	CF	8.101 ± 0.987	0.470	31.0 ± 1.68	4.45 ± 0.39
	AWD	8.124 ± 0.920	0.534	31.0 ± 0.53	5.28 ± 0.77
NG	CF	5.993 ± 1.264	0.436	32.0 ± 1.94	4.12 ± 1.03
	AWD	5.659 ± 0.298	0.452	30.8 ± 0.17	3.65 ± 0.30

Y—Yield (t whole rice grain, 14% of humidity/ha); WP—Water Productivity (Y(kg/ha)/(I + P, m^3^/ha) (kg/m^3^); G—Weight of 1000 grains, with 14% of humidity (g); RS—Rice Straw (dry matter, t/ha); CF—Continuous flooding; AWD—Alternate Wetting and Drying; Experimental sites: BB, Bico-da-Barca; QC, Quinta-do-Canal; and NG, Nuno-Guilherme. Adapted with permission from Oliveira et al. [48], 2022, IGI Global.

**Table 8 sensors-22-03632-t008:** Data and parameters for the automation of farm scale AWD irrigation.

Data Type	Parameters	Description or Comments
General data and rules	Crop database	Development phases duration, Crop coefficients
	Land preparation	Dry or wet sowing, procedure of first flooding
	FML thresholds	Maximum and minimum soil water level (on water tubes), according to the crop phase
	AWD management	When to apply AWD
		Maximum number of days with dry soil per cycle
	Critical weather data	Wind, Temperature
	Actual crop data	Sowing, Pesticide, and Fertilizers application
		Critical water stress phases
Real-time data based on sensors	WFL	Measured in water tubes
	Inflow rate	Measured in weirs, flumes, or water meters
	Meteorological	Temperature, Precipitation, ETo, and wind
		Weather forecast (5–10 days)
Farmer operative rules	Application periods	Dates to apply AWD
	Max. TAD	For crop safety reasons
	FWL thresholds	Water level thresholds on water tubes
Irrigation operation	Inflow gate	Operative mode: on/off
	Outflow gate	Operative mode: on/off
	Field water balance	Get irrigation decision and assess its performance

**Table 9 sensors-22-03632-t009:** Observed Effect Size (ES) of AWD on yield, water use, and water productivity.

Experimental Site	Effect Size (ES) * (%)		
	Y	WU	WP
QC	−3.5	−12.4	10.1
BB	0.3	−11.8	12.8
NG	−5.7	−9.4	3.6

* Calculated by Equation (3); Y—Yield (t whole rice grain, 14% of humidity/ha); WU—Water Use (m^3^/ha); WP—Water Productivity (Y(kg/ha)/(I + P, m^3^/ha) (kg/m^3^); Experimental sites: BB, Bico-da-Barca; QC, Quinta-do-Canal; and NG, Nuno-Guilherme.

## Data Availability

Not applicable.

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
