# Peer review of "Alternate Wetting and Drying in the Center of Portugal: Effects on Water and Rice Productivity and Contribution to Development"

_sensors, 2022, doi:10.3390/s22103632_

Round 1
Reviewer 1 Report
The manuscript proposes an experimental study of alternate wetting and drying flooding in farmer’s paddies by means of sensors’ field irrigation evaluation.
The manuscript is well written and organized.
Research motivations are clearly exposed as well as the research approach.
Also, the results are satisfactorily presented and discussed.
One minor concern is related to Figures 5 and 8, which lack clarity, and need to be improved, e.g. splitting them.
Author Response
One minor concern is related to Figures 5 and 8, which lack clarity, and need to be improved, e.g. splitting them.
Answer:
Figure 5 - The picture quality was improved, through the uniformization of the font and size of the text. Although the figure seems quite heavy, the objective was to superimpose the daily effective temperature recorded at the site with the critical temperatures known for the phenological stages of rice (information from the literature), to facilitate the perception of the phases of the rice cycle in which it was subject to greater thermal stress.
Figure 8 – This graph is an adaptation of Figure 4 by Carrijo et al. (2017) (reference 25), which is more complex as it also considers, in addition to AWD-mild, the AWD-severe. This graph grouping in a single graph of the results for the three soil parameters - pH, SOC, Clayed - allows the comparison between them, which in our opinion is the great advantage in keeping, versus subdividing them into three graphs. We are of the opinion that it is not advantageous to change this figure.
Reviewer 2 Report
Review comments:
Rice is the main food crop in the world, and the realization of precision agricultural management for rice is of great significance to ensure global food security. However, traditional rice irrigation uses flood irrigation (continuous flooding, CF), which has the disadvantage of high water demand and can lead to pesticide pollution of water resources and massive methane emission from soil into the atmosphere. To alleviate this problem, the authors used AWD for the first time in rice irrigation in Portugal. Compared with flood irrigation (continuous flooding, CF), this method saves water resource and improves water resource productivity. At the same time, the new method also reduced rice yields, although only slightly. Overall, it was an interesting and challenging job.
Major questions:
AWD method proposed in this study can effectively improve water resource productivity and realize the purpose of saving water. The authors emphasize that the effect of this approach on rice yield reduction was small, and even the yield increase was slight in the BB study area. According to past experience, AWD method can reduce rice yield. Will the effects of other factors on yield change the current conclusions?
For example, Line 346-347: "The observed effects relative to the soil properties instruct AWD (pH, SOC and texture), are very close to the Carrijo et al. [25] ones obtained worldwide, showing that the effectiveness of the AWD is dependent on those soil characteristics (Figure 7)." Please add how these soil characteristics affect the effectiveness of AWD? Which factors have a greater impact on the effectiveness of AWD? Under what conditions can AWD not be used? Please explain. All in all, this is a very meaningful study. It is hoped that the new method can be validated in more detail in different regions and under different soil characteristics.
Specific modification suggestions:
- Improve the picture quality of Figure 4 and Figure 5. The font and size of the picture should be as uniform as possible.
- Table 9 and Figure 7 have similar meanings. Is it necessary to express the same phenomenon with tables and pictures at the same time?
- Line 329: "I) on yield - the values of QC and NG sites are identical, unlike those from BB, that has an effect very close to zero; " Please revise the statement to make it more accurate. The data parameters of QC and NG are different.
Author Response
Major questions:
AWD method proposed in this study can effectively improve water resource productivity and realize the purpose of saving water. The authors emphasize that the effect of this approach on rice yield reduction was small, and even the yield increase was slight in the BB study area. According to past experience, AWD method can reduce rice yield. Will the effects of other factors on yield change the current conclusions?
Answer:
At each site, the comparison between CF and AWD was made in two plots conducted under identical conditions, so that the only discriminating factor was the irrigation management. Yet, it should be noted that the differences between sites are much greater than between CF and AWD, which demonstrates that soil and microclimate factors are very relevant, as at all sites the same variety and equivalent agronomic practices were used. For the same variety of rice, in a specific site, the main factors that determine the effectiveness of AWD in terms of water savings and the loss of production are, on the one hand, the defining parameters of AWD (timing and thresholds), and on the other hand, the characteristics of the soil, expressed by pH, organic matter, and clay content, that affect the chemical environment of the rhizosphere.
For example, Line 346-347: "The observed effects relative to the soil properties instruct AWD (pH, SOC and texture), are very close to the Carrijo et al. [25] ones obtained worldwide, showing that the effectiveness of the AWD is dependent on those soil characteristics (Figure 7)."
Please add how these soil characteristics affect the effectiveness of AWD?
Which factors have a greater impact on the effectiveness of AWD?
Under what conditions can AWD not be used? Please explain.
All in all, this is a very meaningful study. It is hoped that the new method can be validated in more detail in different regions and under different soil characteristics.
Question: How these soil characteristics affect the effectiveness of AWD?
Answer: To consider this aspect, the following explanation was inserted in the manuscript in section 4. Discussion, after the line 346.
The effect of soil characteristics (pH, SOC and texture) on the effectiveness of AWD in terms of crop productivity could be explained by the following reasons, which relevance is more significant in severe AWD [25,30]: a) Effects of pH: i) on alkaline soils, the high percentage of exchangeable sodium (Na) damages the soil structure and induces impermeable layers; the soil compaction does not have a negative impact on the CF, as the root system is very superficial, but on the contrary, it affects the rice irrigated with an AWD system, as it limits the depth of the roots, which cannot compensate for the lower moisture in the surface soil layers; ii) the high levels of Na in alkaline soils can lead to Na toxicity effects in plants; this problem is mitigated in the CF in the soil saturation condition, due to the higher Na soil leaching and the higher SWP value; however, in unsaturated soil the higher Na concentration causes a greater toxicity risk, thus, rice becomes less tolerant under AWD, with lower productivity; iii) alkaline soils may have higher nitrogen losses due to ammonia volatilization, which is more prevalent in non-flooded conditions; therefore, lower yields under AWD may be due to reduced N root uptake in alkaline soils. b) Effect of SOC: i) the higher soil organic matter affects positively soil water holding capacity and the plant available water, due to its direct impacts on lower bulk density, aggregate stability, high porosity, and improved structure; ii) the higher soil organic matter affects positively the nitrogen mineralization in aerobic soil conditions, implying that N availability may be increased under AWD systems conducted in high SOC soils. c) Effects of texture: the higher clay content reduces the soil percolation, and nitrate leaching, implying that higher levels of nitrogen are available to the root system.
Question: Which factors have a greater impact on the effectiveness of AWD?
Answer:
The major factor impacting on the effectiveness of AWD is the level of AWD severity, that depends on the timing and thresholds set. The soil pH, SOC and texture (clay content) also impact, particularly in the severe AWD option; however, the respective order of importance is not well known, due to the combined complex effects on the soil and the sensitivity of the crop during the dry periods of AWD.
Question: Under what conditions can AWD not be used?
Answer: To consider this question, the following explanation was also inserted in the manuscript in section 4. Discussion, after the line 346.
AWD could not be used in lowland soils with serious drainage problems, where dry soil conditions are not feasible due to the high level of groundwater table (data not shown). On the other hand, AWD is not recommended in situations of high level of salinization of the very shallow groundwater table, to avoid the capillary rise of salts to rice root zone during the dry periods.
Specific modification suggestions:
- Improve the picture quality of Figure 4 and Figure 5. The font and size of the picture should be as uniform as possible.
Answer: These pictures quality was improved, through the uniformization of the font and size of the text.
- Table 9 and Figure 7 have similar meanings. Is it necessary to express the same phenomenon with tables and pictures at the same time?
Answer: Table 9 presents the data obtained in our study, i.e., the effect size of AWD on yield, water use and water productivity) with numerical accuracy. In fact, these data are repeated in Figure 7. Nonetheless, the objective of Figure 7 is to graphically compare those values with the data published by Carrijo et al. (2017), whose data are not presented in Table 9. In our opinion, using both Table 9 and Figure 7, although containing redundant information, allows the reader to have a clearer information than removing Table 9 and using only Figure 7, which only allowed the graphic estimation of our observed data.
- Line 329: "I) on yield - the values of QC and NG sites are identical, unlike those from BB, that has an effect very close to zero; " Please revise the statement to make it more accurate. The data parameters of QC and NG are different.
Answer: This sentence was rectified: "I) on yield - the values of QC and NG sites are both negative (-3.5% and -5.7%), unlike those from BB, that has an effect very close to zero;"
Reviewer 3 Report
There is no reference to figure 1b in the text.
Table 2: a bit unreadable, please consider a different layout e.g. locations in the 1st column, not the header.
Figure 2: for what year or range of years are these values given? This information should be given.
I have no further comments. The paper is well structured, legible. I accept for printing.
Author Response
There is no reference to figure 1b in the text.
Answer: The reference to Figure 1b is in page 4, line149.
Table 2: a bit unreadable, please consider a different layout e.g. locations in the 1st column, not the header.
Answer: To improve the readability, the header of first column was inserted: “Characteristics”
Figure 2: for what year or range of years are these values given? This information should be given.
Answer: The phrase “; data relative to the 1981-2010 period” was inserted in the figure caption.
Round 2
Reviewer 2 Report
I believe the manuscript has been sufficiently improved to warrant publication in Sensors